# Neural Network Structure Optimization by Simulated Annealing

**DOI:** 10.3390/e24030348

**Published:** 2022-02-28

**Authors:** Chun Lin Kuo, Ercan Engin Kuruoglu, Wai Kin Victor Chan

**Affiliations:** Tsinghua-Berkeley Shenzhen Institute, Nanshan, Shenzhen 518071, China; guojl19@mails.tsinghua.edu.cn (C.L.K.); chanw@sz.tsinghua.edu.cn (W.K.V.C.)

**Keywords:** neural network, pruning, structure optimization, heuristics, simulated annealing

## Abstract

A critical problem in large neural networks is over parameterization with a large number of weight parameters, which limits their use on edge devices due to prohibitive computational power and memory/storage requirements. To make neural networks more practical on edge devices and real-time industrial applications, they need to be compressed in advance. Since edge devices cannot train or access trained networks when internet resources are scarce, the preloading of smaller networks is essential. Various works in the literature have shown that the redundant branches can be pruned strategically in a fully connected network without sacrificing the performance significantly. However, majority of these methodologies need high computational resources to integrate weight training via the back-propagation algorithm during the process of network compression. In this work, we draw attention to the optimization of the network structure for preserving performance despite compression by pruning aggressively. The structure optimization is performed using the simulated annealing algorithm only, without utilizing back-propagation for branch weight training. Being a heuristic-based, non-convex optimization method, simulated annealing provides a globally near-optimal solution to this NP-hard problem for a given percentage of branch pruning. Our simulation results have shown that simulated annealing can significantly reduce the complexity of a fully connected network while maintaining the performance without the help of back-propagation.

## 1. Introduction

The successful development of artificial intelligence methods fueled by deep-learning has made an important impact in multiple industries. Elaborate AI algorithms can empower electronic devices to make smart decisions or foresighted predictions [1,2]. However, the outstanding outcomes come with their own cost, which is computational complexity in the case of deep-learning methods. In order to obtain a model that can accurately predict the results corresponding to the input data, a neural network should go through training, which involves initialization [3,4], back-propagation [5], and gradients being updated step-by-step [6,7,8], before use in the inference process. When the dimension of the deep learning structure (layers, number of nodes, branches) is increased, more and more parameters and operations are involved in these four steps.

Neural network compression is a relatively new area of research necessitated by the booming development of deep neural networks and the increasing computational complexity of machine learning methods. Attracted by the extraordinary performance of recently proposed neural network architectures, there are increasingly more demands on edge devices which possess only limited computational resources. Therefore, various network compression techniques have been tried and studied during these years. The recent mainstream strategies can be classified in mainly four approaches: (i) methods that aim at reducing the storage requirements via reducing the number of bits for representing the branch weights that indirectly lead to the reduction of computational load as well, such as quantization [9]; (ii) methods that aim to reduce the computational load by the decomposition of layers or simplifying activation functions, as in [10]. (iii) methods that aim to replace bigger networks with smaller networks that provide similar results for a chosen sub-task, as in knowledge distillation [11]; and (iv) methods that aim to reduce the number of parameters by increasing the sparsity of the neural network, as in pruning [12,13].

In this work, we concentrate on this last class of network compressors and we propose a new method for network structure optimization on the micro scale, differing our work from other works [14,15] that try to optimize the performance (and not necessarily compress the network) by deciding on the macro architectures.

The main idea of pruning is to cut off branches or nodes that “seem” to be “less important”. This is generally based on the weight values. Lower weight values are generally interpreted in the literature as of relatively less importance in a network. Those values that are below a preset threshold are selected as the smallest-K-weight branches to prune from the network [16]. Our first criticism is at this point. We do not share this view that smaller weights are necessarily less important. The simplest way to see it is that smaller weights can lead to branches with larger weights. Even if not so, the highly nonlinear nature of neural networks, makes it difficult to judge the importance of a branch from its weight only. Another important point regarding the achievement of low-complexity neural networks with only pruning lies in the observation that among the training steps of deep learners mentioned above, back-propagation and gradients update make up the major part that consumes the most computing resources and that most of the existing methods still include these two steps in the pruning process.

The direct pruning methods mentioned in the paper [17] propose a proximal gradient algorithm to gradually makes the network sparser with a small learning rate. Another deterministic approximation to binary gating of element-wise parameters [18] also needs gradient descent to minimize the predictions of a targeted network. Ref. [19] prunes the gradients that are smaller than a threshold, which significantly reduces the usage of hardware resources and accelerates the whole training procedure. Another gradient-based approach prunes the weights that are updated toward opposite magnitude [20], providing a chance to reserve those weights with smaller values. These approaches can only be executed on a powerful computer. In addition, if we simply zero out the selected weight values, the network would still occupy a position in a computer represented by zero and, although computational complexity is reduced, the storage requirements would not change. In order to actually reduce the total size of a model while doing inference in a mobile electronic device, various references found in the survey [21] focus on pruning the weight parameters directly instead of the gradients. These pruning techniques can be element-wise, vector-wise, or block-wise, which can correspond to unstructured or structured pruning.

Since a neural network is composed of both nodes and edges, it would be intuitive to set up the optimization problem in terms of the edge weights, the structure of the network, and the pruning strategy. Based on the hierarchical characteristic of the neural network structure, pruning techniques can further be split into both structured [22,23] and unstructured [24,25] pruning categories. When a neural network is simulated on a computer, the majority of computation would be carried out in matrix operations including addition, multiplication, and dot product. In the case of unstructured pruning, the connections between nodes are controlled by an extra mask matrix. When the pruning process is finished, the mask will be mapped back to the weight matrix, replacing some irrelevant values with zeros. On the contrary, the structured pruning strategies sacrifice some optimality to ensure dimension reduction and avoid sparse but high dimensional matrices. The most representative cases for both unstructured and structured pruning are edge-level [26,27] and node-level [28] pruning strategies. If one edge of a network is considered as the minimum unit to reduce configuration, it would not affect the existence of all other elements in the network. However, if one node is the minimum unit to be reduced in a network, all other edges connected to the selected node will also need to be erased.

In choosing pruning strategies, the main factor is to determine which part of the minimum unit (edges, nodes, … etc.) is more important than the others. The existing pruning approaches contain two disadvantages. Firstly, they do not emphasize savings in the consumption of computational resources. Secondly, they do not consider the optimality of the resulting network structure in any means. The optimality under question is determined by two groups of variables/parameters: the branch weights and the branch connections. In this work, we aim to optimize the branch connections with the constraint of a pruning percentage; we augment the network pruning operation with a structure optimization algorithm. For this task, we propose the simulated annealing (SA) algorithm. By considering the sequence of permutations of the pruned configurations as a finite Markov chain, the best remaining configuration can be cleverly searched by simulated annealing learning more and more about the cost surface at each step and eventually arriving at the stationary distribution that coincides with the optimal solution. Different from the other pruning approaches that define additional objective functions and constraints, our approach directly takes the loss function of a neural network as an objective function. The optimized network is guaranteed to be the best permuted configuration under the fixed parameters. To better explain the theory of our proposed method, the needed mathematical symbols which are about to be used in the following paragraphs are included in Table 1.

## 2. Related Works

With the increasing applications of deep learning, there are more and more demands for the portability of the deep neural networks (DNNs) and two lines of research evolved over the years to reduce the storage requirements and computational costs of DNNs. The first line of research focused on reducing the storage requirements of DNNs, which is achieved by formulating the problem as source coding or a data compression problem and making use of existing quantization methods. Examples of this approach include: low bit quantization [29], multi-level quantization [30] and clustering-based quantization [31]. More elaborate source coding schemes are also employed, such as Huffman encoding in [16] and a trellis coded quantization [32] and entropy based coding [33]. With the help of quantization methods, each parameter that occupied 32 bits originally can be reduced to 16 bits or even less bits in a computer without losing network performance.

The other line of approach aims at reducing the number of parameters in the neural network by making it more sparse which is achieved by pruning. Except for the edge-level [26,27] and node-level [28] pruning that are usually applied in a fully connected network, channel-level [34] and filter-level [35,36] pruning were also developed in a convolutional neural network. The pruning approach was motivated by the very early (pre-deep learning) works of LeCun et al. [37] and Hassibi and Stork [38], which showed that neural networks can still work well when many edges or nodes are pruned.

### 2.1. Threshold Pruning Techniques

As exemplified in the paper deep compression [16], one simple yet effective pruning criteria is to select those weights with absolute magnitude smaller than a threshold. The process is executed based on a pretrained network. Once the small weights are pruned, the remaining weights will be fine-tuned by training with the dataset again. Various pruning algorithms differ in their choice of where they apply the threshold, such as the output of activation function or the gradient of each weight parameter or the magnitude of the weights. It has also been reported in the deep compression approach [16] that pruning the network gradually, by picking up only a small fraction of total weights parameters at one time can help to maintain the performance of the pruned network. However, the sparse network still occupies a lot of memory in a computer because the pruned weights are all represented by zeros. In order to truly eliminate the parameters from the network, [39] proposed tight compression which converts the resulting large sparse matrix after pruning to a smaller but dense matrix by moving non-zero members to matrix locations with zeros. These moves are not done in a deterministic manner but instead via a stochastic heuristic algorithm, namel, y simulated annealing. In our work, magnitude threshold pruning is considered as our baseline. The parameters with top K minimum absolute values will be pruned which is called min K strategy.

### 2.2. Gradient Pruning Techniques

Ref. [20] states that the small weights do not necessarily mean they are useless. A branch with a small magnitude weight value would still be considered as an important branch if it has a relatively large magnitude gradient. The existing pruning techniques prune the network either during the training process, or after the training process is completed. However, there is a huge difference of computational criteria between training a network and simply doing inference using a network. In order to reduce the computational cost, ref. [19] proposed to prune the gradients propagated from the activation function under a threshold. By removing the gradients that are close to zero, the training speed can be faster and the computation cost can be reduced. But this approach still uses back propagation during the pruning process.

### 2.3. Other Annealing Applications

A small number of researchers have applied simulated annealing algorithm to the field of deep neural networks in recent years. However, these works focus on improving the training process of a network. Ref. [40] claims that SA can be used to update the parameters of a convolutional neural network. Ref. [41] applies SA to find the optimal initialization based on the prior extracted features from an image dataset. Ref. [42] integrates a modified annealing algorithm to the weight training process to achieve higher accuracy. Different from these papers, which utilize SA for weight optimization, we utilize SA to estimate optimal network configuration that leads to best accuracy. Our results show that SA is capable of finding the more important parameters so that the less important parts can be pruned effectively.

### 2.4. Other Heuristic Algorithms

Other than simulated annealing, particle swarm optimization (PSO) and genetic algorithms (GA) are also included in multiple previous works [43,44,45,46], which search for an optimal solution by iteratively improving the solution via intelligent searching strategies. The main difference between these methods is the theory that support their convergence. SA is guaranteed to get the global optimal solution with the condition of detailed balance following the Markov chain theory. On the other hand, PSO and GA risk falling into local minima. A more detailed comparison can be part of future works to evaluate the advantages between meta-heuristic algorithms.

## 3. Network Optimization Using Simulated Annealing

In this work, we propose a heuristic, non-convex optimization algorithm, namely simulated annealing (SA), for the structure optimization of partially connected neural networks after pruning [47]. The choice of simulated annealing has been motivated by the success of the algorithm in various problems involving network/graph structures with a large number of configurations and complicated cost surfaces with various local minima [48,49,50].

SA is motivated by the annealing process in solid state physics, which aims to place the electrons in a solid at their lowest energy states, achieving the lowest possible energy configuration [51]. In the solid state annealing process, the solids are heated to a high temperature in which all electrons are basically free to move to any energy configuration, even to higher energy ones due to having high thermal energy. The solid is then cooled slowly, allowing time for the electrons to settle to the lowest energy configurations. This scheme allows the electrons to avoid local optimal configurations. SA simulates this scheme almost exactly: the solution space is explored with a random walk that moves through neighboring configurations rather than randomly picking solutions. This random walk is also a special one; the transitions between solution states in the steps of the random walk is dictated by a Markov chain. At each step of the random walk, the steps are accepted or rejected according to a Boltzmann statistic calculated over the cost function of the new and the old configurations as in the case of electrons in a solid. This accept/reject mechanism ensures that the Markov chain has detailed balance property and, hence, is irreducible and aperiodic, which means that it has a stationary limiting distribution. The reduction of temperature artificially in SA via scaling the accept/reject probability enables the Markov chain to converge to an increasingly peaked stationary distribution and the peak of the distribution gives the optimal solution to the optimization problem.

Similar to typical discrete optimization problems, SA is given a finite set containing all possible configurations C—in our problem, a fully connected neural network N(·) and a loss function L(·)—and looks for $c^{*}\in C$ such that $L(c^{*})$ is minimized. The loss function defined in the optimization problem of neural network is in analogy to the energy function defined in solid state physics. Different from the panorama of gradient descent algorithms, which are very likely to be trapped in local optima, simulated annealing algorithms avoids the local optima by allowing the acceptance of worse off solutions according to the Boltzmann dynamics [47].

Simulated annealing is a general algorithm that can be applied to various problems. In order to solve a specific problem, SA requires some customization of its components by the user. The success of SA depends on the careful design of three crucial mechanisms: state neighborhood structure selection, acceptance-rejection criteria of proposals, and the cooling schedule.

### 3.1. Choice of State Neighborhood Structure

The choice of neighborhood structure dictates the possible moves the random walk can make; hence it affects the convergence rate significantly. A too-conservative neighborhood structure would make the exploration of the solution space very slow while a too-liberal choice may make the random walk jump over important minima and will not learn enough, turning it into a blind random search algorithm on all possible configurations. However, a conservative or liberal neighborhood structure does not always provide the same level of disadvantage or advantage. At the beginning of the annealing process, big changes between neighborhoods can be helpful for a quick and coarse search of the solution space. When a potentially interesting part of the solution space has been reached, smaller moves in a conservatively defined neighborhood would be more beneficial; there is more space being looked around, small changes turn out to be more beneficial to capture the global optimum. On the other hand, starting with too-conservative neighborhoods leaves the algorithm stuck in local minima near the starting configuration.

For the network structure optimization problem, we define the states in the solution space as the configuration of branches connecting nodes in one layer to the next one. Equivalently, we consider a mask matrix $M_{\ell }$ that is filled with ones at locations (i,j), where there is a surviving branch between the nodes *i* and *j*. After pruning weights at *ℓ*-th layer, ones will be replaced by zeroes at locations (i,j) where there are no surviving branches. The meaning of these matrices’ operation can be visualized in Figure 1, where the colorful edges shown in left part representing the surviving branches.

The neighboring states are defined to be new configurations $M_{\ell }^{\prime }$ obtained by moving one (or more) branch(es) originally at location (i,j) to (i,k). The changes in the mask matrix is equivalent to the position changes to the zero elements in Figure 1 from matrix location (i,j) to another position on the same row (i,k). The choice of a single branch replacement corresponds to a conservative neighborhood structure. It can be made more liberal by considering more than one replacement.

### 3.2. Acceptance–Rejection Ratio

The simulated annealing algorithm can jump out of a local optimum, since it occasionally accepts a new state that increases the outcome of the loss function L(·). This mechanism reflects an analogy with the electron dynamics in solid state physics; when the electron has enough thermal energy it can jump over barriers and end up in a higher energy state and hence can avoid some local energy minima. As in the case in electron dynamics, SA adopts a Boltzmann distribution in deciding to accept or reject a move or a step of the random walk:(1)$$\begin{array}{c} P_{b}=\min\left(1,\exp\left(\frac{-\Delta L(\cdot )}{k\cdot T}\right)\right) \end{array}$$

At every move, $-\Delta L\left(\cdot \right)=L\left(M\right)-L\left(M^{\prime }\right)$ is calculated; if it is larger than zero, the new state is accepted, otherwise a uniform number is generated between [0,1) and compared with $P_{b}$. If $P_{b}$ is greater it is accepted, otherwise it is rejected.

### 3.3. Convergence

An important question is whether this accept/reject scheme and the choice of neighborhood structure ensures convergence. The answer to this question is well known in the Markov chain theory literature [52]. A (finite-state) Markov chain converges to a unique stationary distribution only if it has two fundamental properties: aperiodicity and irreducibility or, equivalently, ergodicity. These properties are not easy to check generally and therefore a stronger property is used, namely detailed balance. For our problem, the detailed balance condition can be expressed as:(2)$$\begin{array}{c} P_{c}\left(M\right)\cdot P\left(M^{\prime }|M\right)=P_{c^{\prime }}\left(M^{\prime }\right)\cdot P\left(M|M^{\prime }\right) \end{array}$$

$P_{c}$ indicates the probability that the configuration is sampled under the representation of mask matrix M, and $M^{\prime }$ indicates the first-order neighbor configuration of the Markov chain after M. It can be shown that the accept/reject mechanism using the Boltzmann function in Equation (Equation 1) ensures that the detailed balance condition holds and therefore converges to the unique stationary distribution [52]. This property is shared with the Markov chain Monte Carlo (MCMC) algorithm, which aims to obtain the posterior distribution of model parameters. Both algorithms construct Metropolis loops via random walks over Markov chains, satisfying the detailed balance condition. The difference is that the simulated annealing algorithm employs several Metropolis loops with a decreasing temperature parameter *T* in Equation (Equation 1). Hence, posterior distribution, approximated by every Metropolis loop, becomes increasingly peaked at the maximum of the stationary distribution. In this way, the parameter values that maximize the stationary distribution of the Markov chains are obtained.

### 3.4. Cooling Scheme and Hyperparameters

Another design issue affecting the convergence rate of SA is the cooling scheme. It is known that the convergence is guaranteed in the case of logarithmic cooling; however, this requires infinitely slow cooling. Instead, most users prefer geometric cooling.
(3)$$\begin{array}{c} T_{n+1}=\eta T_{n}. \end{array}$$

The important parameters to be set are the initial temperature ($T_{init}$), the cooling rate (η) between metropolis loops, and the metropolis loop length (MLL). The temperature decrease controls the acceptance–rejection probability for a new state that has higher loss value. If the temperature is high, the worse states are more likely to be accepted. As the temperature is reduced increasingly, the worse states are more likely to be rejected. The decreasing rate is the main factor responsible for the speed of the annealing process. However, a faster annealing process does not guarantee the convergence. The objective function can be minimized only when the decreasing speed is slow enough. In our work, we setup the initial *T* as 10 and η as 0.98. The third parameter MLL should be large enough for the Markov chain to converge to the stationary distribution and small enough not lose time unnecessarily beyond guaranteeing convergence.

### 3.5. Selection of Weight Parameters

Simulated annealing is good at finding the best permutation from a finite set given a number of total selections. The important elements among those encountered during the random walk are determined by the Boltzmann criterion. Each selection decision can be formulated as a discrete number {0,1} where zero and one indicate that the elements are ignored and selected respectively. These two types of values are actually the fundamental elements composing the mask matrix (*M*). During the annealing process, the change of selections indicates the position exchange between a pair of zeros and ones. The details are described in Algorithm 1 where a network takes a dataset T and a set of mask matrices as inputs. The *c* marked in the bracket indicates the *c-th* type of configuration represented by *M*. The mask matrix at different layers should strictly follow the number of input and output node of the layer where the mask matrix belongs to.
**Algorithm 1** Selection of Weight Parameters using SA**Input:**$N\left(T_{train},{\left\{M_{\ell }^{\left(c\right)}\right\}}_{\ell =1}^{k}\right),M_{\ell }^{\left(c\right)}={\left\{0,1\right\}}^{in\times out},c\in C$**Parameter:**$T_{init}$, $T_{min}$, η, *k*, MLL**Notes:**MLL stands for “Metropolis-loop length”**Output:**$N(T_{train},{\left\{M_{\ell }^{\left(c^{*}\right)}\right\}}_{\ell =1}^{k})$1:Let $t=T_{init}$.2:**while**$t>T_{min}$**do**3:   **for** i in 1∼MLL **do**4:     Compute $Loss=L(N,T_{train}^{labels})$5:     Randomly select one connected link in hidden layer.6:     Randomly select one disconnected link in hidden layer.7:     Switch the connection status of two selected links $M_{hidden}^{\left(c^{\prime }\right)}\leftarrow M_{hidden}^{\left(c\right)}$8:     Compute $Loss^{\prime }=L\left(N^{\prime }\left(M_{hidden}^{\left(c^{\prime }\right)}\right),T_{train}^{labels}\right)$9:     **if** $Loss^{\prime }<Loss$ **then**10:        Accept new configuration $N\leftarrow N^{\prime }$11:     **else**12:        Random r(0,1)13:        **if** $r>\exp\frac{-(Loss^{\prime }-Loss)}{k\cdot T}$ **then**14:          Accept new configuration $N\leftarrow N^{\prime }$15:        **else**16:          Remain in the original network configuration17:        **end if**18:     **end if**19:   **end for**20:   t←η·t21:**end while**22:**return**$N(T_{train},{\left\{M_{\ell }^{\left(c^{*}\right)}\right\}}_{\ell =1}^{k})$

### 3.6. Permutation after Edge Pruning

According to Algorithm 1, the mechanism starts by setting how many less important edges to eliminate. Even though the elements might be eliminated during the random walk, they still have a chance to be recovered back. Starting from a sparser network, an optimal sub-configuration can be determined iteratively by simulated annealing algorithm. If the total number of elements is larger, the corresponding Markov chain must be longer so that a wider space is explored. The longer length indicates that the initial temperature should be higher and the decreasing rate should be closer to one so that SA has more time to search for the whole space.
(4)$$\begin{array}{c} \underset{w_{\ell }^{*},b_{\ell }^{*}}{min}L\left(N(T_{train}|{\left\{w_{\ell },b_{\ell }\right\}}_{\ell =1}^{k},T_{train}^{labels}\right) \end{array}$$

Since the network performance will strictly go down as increasingly more parameters are erased, the optimal permutation is determined under the given pruning percentage. Different from [40] which applied SA to update weights directly, we implemented it to decide which edges should be kept connected and which should not be in each iteration. In order to change connections in a more convenient way, the actual objects that we are fine-tuning would be the mask matrices, which is in accord with Figure 2. The whole optimization process starts from a well trained fully connected network, which is done by gradient descent repeatedly. In order to avoid unexpected potential interference to the experimental results, we design a simple network that contains only four layers, since it is easier to clarify the effectiveness and contribution of a proposed optimization method. The full structure is illustrated in Figure 3. After obtaining the weight parameters for the given percentage of pruned branches, we erase a fraction of total weight parameters and start the annealing process. These two steps can both be fulfilled by manipulating the mask matrices.

According to the pruning objective, which is minimizing the same loss function L(·) applied in Equation (Equation 4) by fine-tuning the network configuration, the objective function can be formulated as follows.
(5)$$\begin{array}{c} \underset{M_{\ell }^{\left(c^{*}\right)}}{argmin}L\left(N\left(T_{train}^{data},{\left\{M_{\ell }^{\left(c\right)}\right\}}_{\ell =1}^{k}\right)|{\left\{w_{\ell }^{*},b_{\ell }^{*}\right\}}_{\ell =1}^{k},T_{train}^{labels}\right) \end{array}$$

The objective function Equation (Equation 5) optimizes the configuration starting from a fixed preset fraction of a randomly pruned network, followed by the steps of Algorithm 1. Our experiment shows that the network performance can be maintained within a reasonable range of pruning fraction without further fine-tuning by back-propagation.

## 4. Experimental Study

According to the pruning strategy introduced above, which is to optimize Equation (Equation 5) by applying Algorithm 1, the main factors that influence the performance of SA on optimized network are three parameters: Metropolis loop length, initial temperature, and temperature decrease rate. In order to evaluate the performance of the pruned network under different conditions, two datasets were used in the experiments individually in this paper, namely MNIST and FASHION.

To see the effect of metropolis length on the performance, Markov chains of various Metropolis loop lengths (MLL∈0,1,10,20,50,100) have been simulated. Longer random walks, as expected, increase the performance of the final result; however, this comes at the cost of increased time complexity of Algorithm 1. Based on our experiments, we conclude that 20–30 MLL have been seen to be good compromise to get both good performance and tolerable time consumption.

### 4.1. Visualization to the Selection of Weight Parameters

In order to observe the performance of a pruned network at different pruning percentages, a fraction of pruning percentage p∈[0,1) is setup to execute each strategy gradually. To compare the improvement between our method and naive pruning approaches, threshold pruning (min-K) strategy [16] is included in the experiment as the baseline. The K minimum absolute weight values will be pruned corresponding to a given fraction. In addition, to acknowledge the differences between a strategic pruning approach and doing nothing, uniformly random pruning is also included.

As the histograms shown in Figure 4 illustrate, the pruning process strictly follows min-K strategy starting from zero. The reason why the distribution of these four histograms are slightly different is because each pruning starts from a new and independent well-trained, fully connected neural network. To guarantee the consistency of our experimental results, each result is the mean value of at least five trials. The other histogram set, which is the four graphs included in Figure 5, shows the pruning process followed by the simulated annealing algorithm. Contrasting the two figures, the trend shows that the importance of a weight parameter is correlated to its magnitude, in the beginning. However, as more and more less-effective weights are pruned, this correlation becomes weaker. As the Metropolis loop length in the annealing process is increased, Figure 4 pruned branches are all focused around the mean value while those in Figure 5 disperse to high-amplitude-weight branches, as well.

### 4.2. Performance Trend under Different Pruning Scales

We have compared the performance from *random*, *min-K*, and our simulated annealing based method at various pruning scales progressively in Figure 6 and Figure 7.

According to the comparison of accuracy values at different percentages of pruning in Figure 6, with simulated annealing (with metropolis length longer than 20), most of the lost accuracy can be recovered until 90% pruning. Beyond the 90% line, despite a drop in accuracy, the accuracy is still significantly higher than those of random pruning and min-K pruning until 99% pruning. In Figure 7 similar patterns are observed, albeit with reduced accuracy values due to having a more complicated dataset. The simulated annealing algorithm shows its potential in finding the optimal configuration when all weight parameters are fixed.

Both MNIST and FASHION are opensource datasets containing more than 50 thousand images related to 10 different classes of object. If the accuracy of a network is close to 10%, it would be equivalent to randomly guessing the results. Our results have shown that the accuracy can be effectively raised by selecting few crucial weight parameters corresponding to the near-optimal pruned network configuration. According to our experiments on two different datasets, pruning followed by SA (Algorithm 1) shows its value especially when there are less than 30% of weight parameters remaining in the network.

Since the objects in the FASHION dataset are more complex, the accuracy predicted by a simple neural network is lower comparing with the performance evaluated in the MNIST dataset. However, by applying our pruning algorithm to a network with less accuracy, the performance of the network can still be better reserved compared with the min-K strategy. It proves that the effectiveness of Algorithm 1 works independent of the network accuracy and the performance degradation can be postponed by applying Algorithm 1. According to the experimental results, it is more secure to set the MLL to 50 due to the complexity of various datasets.

On the other hand, the disadvantage of SA is the restriction caused by fixed weight parameters. Although the permutation can determine the importance of various weights, the values are only updated when the neural network is trained by gradient descent. All of the steps done in Algorithm 1 are only related to network permutation by selecting optimal branch connections for the given set of weights.

The experiments on the two datasets in Figure 6, Figure 7, Figure 8 and Figure 9 have shown that SA can recover most of the performance lost by pruning. For example, when 90 % pruned the network loses more than 20% and SA recovers more than 15% of the lost performance and reduces the performance loss to around 4%. For certain datasets and pruning ratios for which pruning leads to less than 0.50 accuracy, SA is capable of increasing the accuracy to above 0.50.

In this work, the feasibility of a new pruning strategy based on the theory of simulated annealing has been demonstrated on a simple network. In the future works, the application of network pruning can be further extended to the deeper and more complex networks.

### 4.3. Time Complexity of the SA-Based Pruning Process

The total time consumption of Algorithm 1 is dictated by three factors: initial temperature ($T_{init}$), temperature update coefficient (η), and total number of temperature updates. In order to find out the best set of parameters, the convergence process should be monitored. If ΔL(·) is a small value, the initial temperature should be lower and the temperature decreasing rate should get closer to one. In our experiment, $T_{init}$ is set as 0.2 and η is set as 0.95. After updating the temperature for approximately 100 to 150 iterations, the network performance is observed to converge. The time consumption of Algorithm 1 for different MLLs (1, 10, 20, 30, 40, 50, 60, 70, 80, 90) are measured for each one of three different fractions of gradual pruning percentage *p* (2%, 5%, 10%) in Figure 10.

Even though the extreme cases are included in the time consumption evaluation; MLLs of 20 to 50 seem to be enough, as can be observed in Figure 7. We have evaluated, also, the impact of network size. Comparing with the regular network training by back propagation, simulated annealing can be done on network structures without computing the gradients, which saves huge memory space for the device. Based on the scenario that the computational power of edge devices is not enough to train a deep neural network, our approach provides a path to pruning weight parameters without further weight training. The implementation of the pruning process is done hierarchically, layer by layer. If there are more layers contained in a deep neural network (DNN), the time complexity will increase linearly. On the other hand, if there are more nodes contained in one layer, the time complexity increases sub-logarithmic with increasing number of branches. When the network size is increased 10 times the time increases 1.2 times and when it is increased by $10^{4}$ times the time increases by 9 times.

The whole experiment and evaluation were run on a Windows computer with an Intel Core i3 1110G4 CPU processor and limited 3GB random access memory (RAM) shared by Nvidia TITAN-RTX GPU. According to the experimental results illustrated in Figure 11, the size of a network influences the total time consumption only fractionally. The main reason for the change in the slope is caused by the hardware parallel computational differences with respect to various network scales. In the case of taking MNIST and FASHION as datasets, there are more similar features being extracted by a bigger network. This makes the probability of getting these close features from a bigger network remains similar to the probability of getting the features from a smaller network. Even though the optimized layer is increased by 10 thousand times, which are 3200 input nodes and 1600 output nodes in the hidden layer, the needed time still increases less than 250 s. By optimizing the networks with different numbers of edges using the same hyperparameters mentioned in the first paragraph of Section 4.3, the final performance proves that they all converge successfully within the same update iteration. In addition, the computational power of the hardware also affects the time consumption of the pruning process. The listed times were counted under the case when all of the testing data can be predicted at once. If the RAM of the edge device can not contain them all, the whole pruning process will take several times longer than the listed ones.

## 5. Discussion

Even though there are limits determined by the training data, the performance of more elaborate networks can go far beyond this level, such as in VGG and ResNet. The key reason for the higher performance of these networks is because the convolution operation can extract special features from image dataset more efficiently. In order to enhance the computation speed to convolution, one paper has suggested a method that can transform the kernel scanning process into matrix multiplication [53], which is equivalent to the operation done in a fully connected network. This implies that the potential performance of a fully connected network is very likely to be discovered by finding better methods for updating weight parameters. The global optimum obtained by simulated annealing is only for the case when the parameters are fixed. By permuting the network configuration with Algorithm 1, the final results can outperform the classical pruning methods that consider magnitude threshold as their pruning criteria. It has also been experimentally shown that the weight parameters with small magnitudes are not certain to be the less important parameters. With the help of SA to select key weight parameters, a more lightweight configuration can be obtained without sacrificing performance.

The other advantage of our work is that the pruning and permuting processes with simulated annealing algorithm involves only forward pass. It indicates that RAM space contained in a computer can be hugely saved, and the pruning process can therefore be executed on a less powerful computer. It indicates that the cloud computing will not be a necessity in order to prune a network. In the era of AI technology, health-care industry can be a good potential application area due to huge amount of data and lack of enough computational resources. Namely, the network inference and training processes are executed in separated devices. In order to implement some intelligent functions in their systems, they usually need to cooperate with cloud providers. Once a well-trained model is provided, the user would not be able to modify it, especially when the computational complexity of the model is high. With the help of our algorithm, a feasible path is provided to run the pruning process locally regardless of cloud providers as long as the device can execute inference process. In addition, our work has strong adaptability and scalability to other types of network pruning. In this paper, we took edges as the basic elements of the configuration, which necessitated long Markov chains. However, redefining the state space and neighboring state structure, simulated annealing algorithm can be coded also for other types of basic elements, such as nodes and convolution filters. By asking SA which filter to erase instead of which edge, the time complexity can be reduced to acceptable levels in the pruning process of complex networks with the help of GPUs.

## Figures and Tables

**Figure 1 entropy-24-00348-f001:**
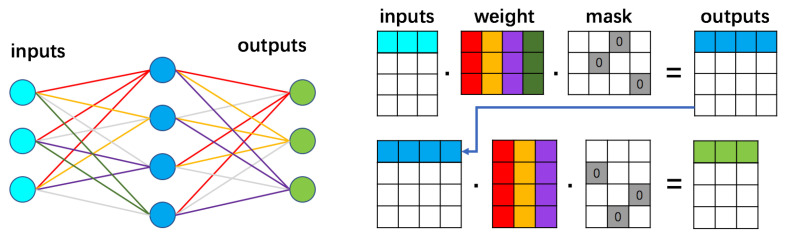
Illustration for the implementation of a neural network in a computer. Each edge is represented by a number located in the weight matrix and the colors categorize where the links are connected to. The status of connection and disconnection is controlled by a mask matrix and the colors of the network illustration strictly follow what is set up in the matrices’ operation.

**Figure 2 entropy-24-00348-f002:**
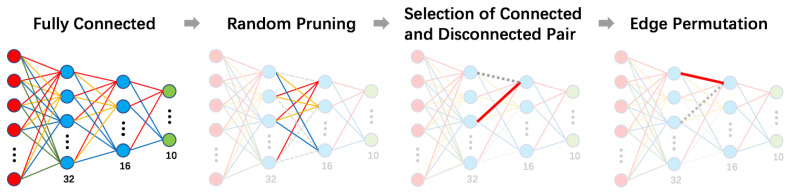
Illustration for our proposed methodology where SA is mainly working on the pair selection and edge permutation parts.

**Figure 3 entropy-24-00348-f003:**
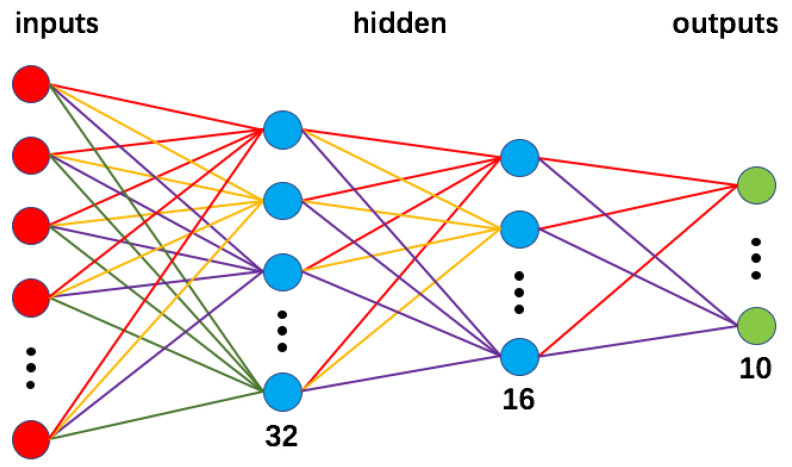
Network structure of a shallow network. This network is designed to evaluate the effectiveness of SA in the field of network pruning. In order to keep things simple, only the edges in hidden layers will be pruned.

**Figure 4 entropy-24-00348-f004:**
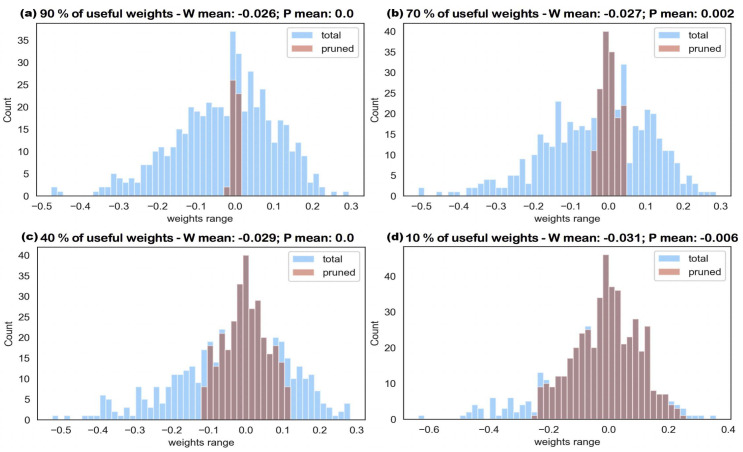
The histogram of total weight parameters under the process of gradual pruning using threshold pruning strategy with pruning percentage p=10%; (**a**) 90% of weight parameters remain; (**b**) 70% of weight parameters remain; (**c**) 40% of weight parameters remain; and (**d**) 10% of weight parameters remain.

**Figure 5 entropy-24-00348-f005:**
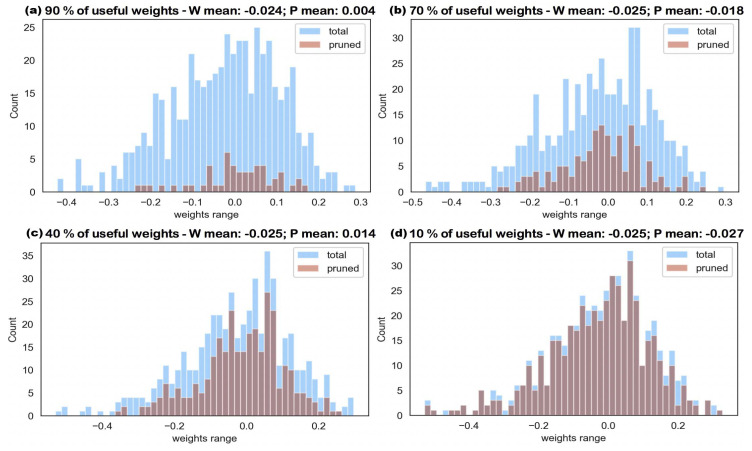
The histogram of total weight parameters under the process of gradual pruning followed by Algorithm 1 with pruning percentage p=10%; (**a**) 90% of weight parameters remain; (**b**) 70% of weight parameters remain; (**c**) 40% of weight parameters remain; and (**d**) 10% of weight parameters remain.

**Figure 6 entropy-24-00348-f006:**
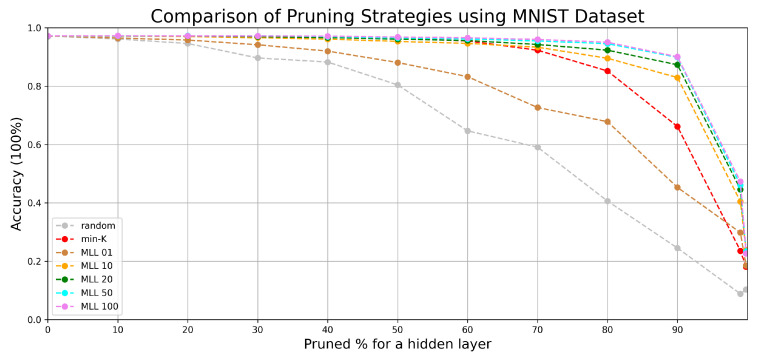
The accuracy decrease corresponding to the pruning scale of the fully connected network that was originally trained by MNIST dataset.

**Figure 7 entropy-24-00348-f007:**
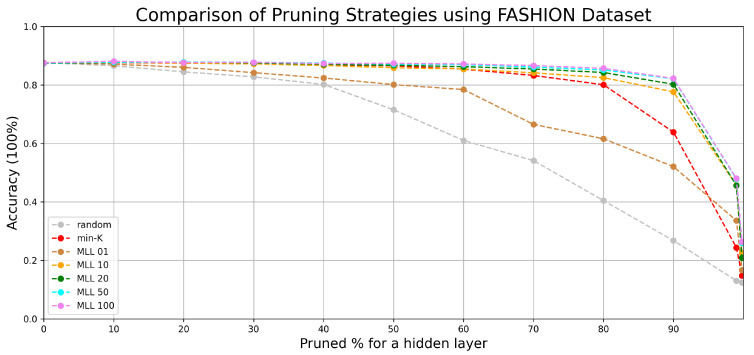
The accuracy decrease corresponding to the pruning scale of the fully connected network that was originally trained by FASHION dataset.

**Figure 8 entropy-24-00348-f008:**
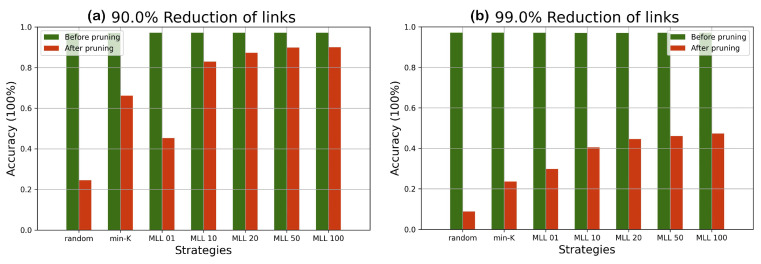
The performance differences before and after pruning the network under different strategies given a predefined pruning percentage on the MNIST dataset; (**a**) 90% of links are pruned; and (**b**) 99% of links are pruned.

**Figure 9 entropy-24-00348-f009:**
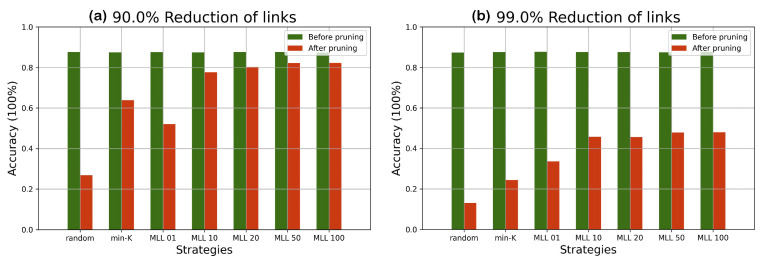
The performance differences before and after pruning the network under different strategies given a predefined pruning percentage FASHION dataset: (**a**) 90% of links are pruned; and (**b**) 99% of links are pruned.

**Figure 10 entropy-24-00348-f010:**
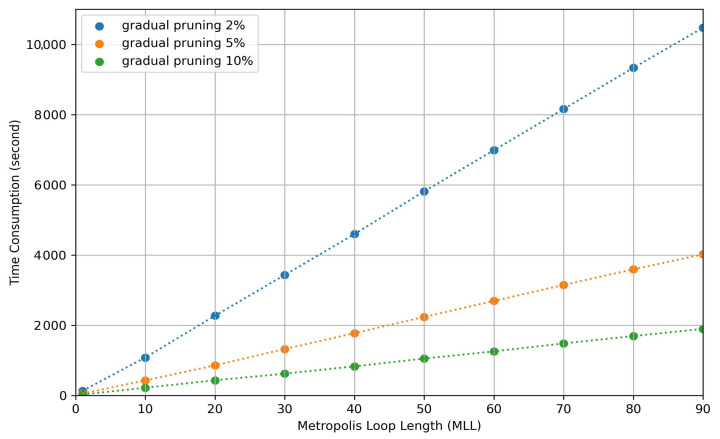
The visualization of time consumption for values of 1, 10, 20, 30, 40, 50, 60, 70, 80 and 90 of metropolis loop length under different pruning percentages.

**Figure 11 entropy-24-00348-f011:**
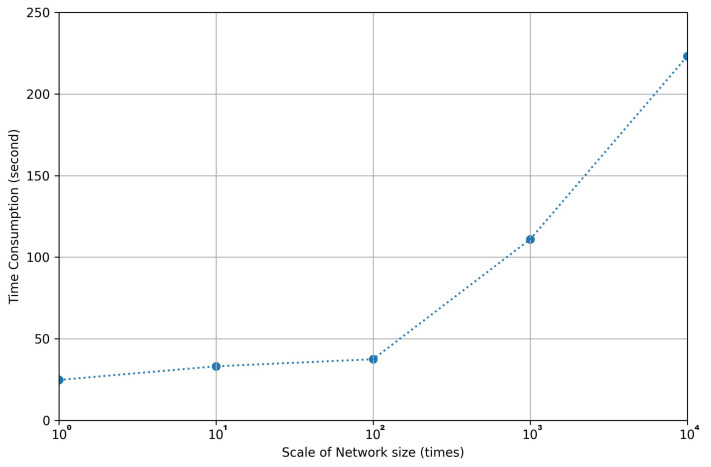
The visualization of time consumption during the one-shot pruning process between 1, 10, $10^{2}$, $10^{3}$ and $10^{4}$ scale of network size. The actual number of edges are 32 × 16, 320 × 16, 320 × 160, 3200 × 160 and 3200 × 1600 respectively.

**Table 1 entropy-24-00348-t001:** Mathematical symbols that are used in this paper.

Symbol	Name	Description
C	configuration set	all possible network configurations
c	one configuration	one of the configuration included in the set
$c^{*}$	optimal configuration	optimal configuration evaluated by a function
L(·)	loss function	a function for performance evaluation
N(·)	network function	a function to formulate a DNN
$P_{b}$	Boltzmann probability	probability function applied in SA
$P_{c}$	configuration probability	probability of getting this configuration
*p*	pruning %	% that the hidden layer of a network is pruned
$M_{\ell }^{\left(c\right)}$	mask	the mask in *ℓ*-th layer representing “c” configuration
$M_{\ell }^{\left(c^{*}\right)}$	optimal mask	the optimal mask in *ℓ*-th layer representing “c” configuration
M	mask set	all mask matrices belonging to the network
$w_{\ell }$	weight matrix	weight matrix belonging to the *ℓ*-th layer.
$w_{\ell }^{*}$	optimal weight matrix	optimal weight matrix belonging to the *ℓ*-th layer.
$b_{\ell }$	bias	bias term belonging to the *ℓ*-th layer.
$b_{\ell }^{*}$	optimal bias	optimal bias term belonging to the *ℓ*-th layer.
*k*	constant	a coefficient to control accept–reject rate
*T*	temperature	the temperature in SA
η	decreasing rate	the coefficient to decrease the temperature
T	dataset	a dataset for network training and testing

## Data Availability

The data presented in this study are available on request from the open source websites. MNIST is available at http://yann.lecun.com/exdb/mnist/ and FASHION is available at kaggle https://www.kaggle.com/zalando-research/fashionmnist, which is widely considered as the primer image dataset for machine learning. Both datasets applied in this paper were downloaded from these two websites on 1 February 2020. The experimental data and code can be reviewed at https://github.com/khle08/Network-pruning-with-Simulated-Annealing (assessed on 19 December 2021). The code is written in Python 3.6 including *torch* (1.6.0), *torchvision* (0.7.0), *numpy* (1.19.5), *pandas* (1.1.5), and *matplotlib* (3.3.3).

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
