# Peer review of "Neural Network Structure Optimization by Simulated Annealing"

_entropy, 2022, doi:10.3390/e24030348_

Round 1
Reviewer 1 Report
The importance of the paper is high but the base for the research should be better described. Also, the state of the art could be better prepared.
The authors use some abbreviations like DNN without explanation, I suppose it means Deep Neural Network, but it should be clear. I suggest adding the expansion of the abbreviation. In the list of abbreviations at the end of the paper is a lack of DNN. Why MDPI, DOAJ, TLA are in the abbreviations?
Chapter 2 there is presented the reducing the storage requirements of DNN and the examples are presented (just mentioned). In the second paragraph, the pruning method is mentioned but without examples. In subchapters 2.1 and 2.2. the methods based on pruning are described. The authors present the inconsistency in the writing. I suggest adding examples of pruning in the second paragraph (before subchapter 2.1). And then add the description methods included in reducing the storage requirements with description (before subchapter 2.1).
The literature should be extended by the papers related to ANN and real cases like Predicting the probability of cargo theft for individual cases in railway transport, Technical Gazette, Vol. 27/No. 3; PickupSimulo–Prototype of Intelligent Software to Support Warehouse Managers Decisions for Product Allocation Problem, Applied Science vol. 10, no. 23.
Sometimes Simulated annealing is written by one high and one low letter, in other cases by two high letters. It must be normalized, I suggest writing it in two high letters.
Not each symbol used in equations is described. I suggest adding the list of symbols at the end of the paper.
Why the number of algorithm 1 in lines 263, 293, 303 is in blue?
Chapter 4.1. starts with a figure. It should start with text not figure.
Figures 5 and 7 should be bigger (the font is readable but hardly).
The reference formatting is not properly prepared. The references should be format regards to the Journal requirements.
Author Response
The authors use some abbreviations like DNN without explanation, I suppose it means Deep Neural Network, but it should be clear. I suggest adding the expansion of the abbreviation. In the list of abbreviations at the end of the paper is a lack of DNN. Why MDPI, DOAJ, TLA are in the abbreviations?
Response 1: Thank you for pointing out this. Open forms of the abbreviations have been added and other irrelevant abbreviations have been removed.
Chapter 2 there is presented the reducing the storage requirements of DNN and the examples are presented (just mentioned). In the second paragraph, the pruning method is mentioned but without examples. In subchapters 2.1 and 2.2. the methods based on pruning are described. The authors present the inconsistency in the writing. I suggest adding examples of pruning in the second paragraph (before subchapter 2.1). And then add the description methods included in reducing the storage requirements with description (before subchapter 2.1).
Response 2: The example of pruning and the description methods of storage requirement have been supplemented in the first and second paragraph of Chapter 2 before subchapter 2.1.
The literature should be extended by the papers related to ANN and real cases like Predicting the probability of cargo theft for individual cases in railway transport, Technical Gazette, Vol. 27/No. 3; PickupSimulo–Prototype of Intelligent Software to Support Warehouse Managers Decisions for Product Allocation Problem, Applied Science vol. 10, no. 23.
Response 3: Thank you very much for drawing our attention to these 2 articles. However, we think they are not related with the subject matter although they could be future potential applications of the methodology we develop.
Sometimes Simulated annealing is written by one high and one low letter, in other cases by two high letters. It must be normalized, I suggest writing it in two high letters.
Response 4: Thanks for the notice. “Simulated Annealing” has been uniformly adopted throughout the paper.
Not each symbol used in equations is described. I suggest adding the list of symbols at the end of the paper.
Response 5: Thanks for your suggestion. A list of symbols has been added in table 1.
Why the number of algorithm 1 in lines 263, 293, 303 is in blue?
Response 6: It is because the algorithm number has been linked to the algorithm steps table with \ref{…}, which is according to the official template of MDPI.
Chapter 4.1. starts with a figure. It should start with text not figure.
Response 7: Chapter 4.1 has been edited properly.
Figures 5 and 7 should be bigger (the font is readable but hardly).
Response 8: Both figures have been enlarged to a proper size.
The reference formatting is not properly prepared. The references should be format regards to the Journal requirements.
Response 9: We have carefully examined the issue of reference formatting. The irregular formats have been corrected.
Reviewer 2 Report
This research article addresses the effect of Simulated Annealing algorithm on large neural networks, thereby the authors conducted globally near-optimal solution to reduce the complexity of a fully connected network. The reviewer recommended its publication by considering minor revisions:
- The literature review in lines 18-27 needs supporting references.
- Line 44: “…other work…” needs a reference. It is not clear here which research work you are referring.
- Lines 75-90 needs references. Otherwise, discussions are not supported
- Line 110: needs to define DNNs stands for what?
- Overall, authors needs to carefully edit their manuscript there are some errors in text, for example line 140, 320,…

Author Response
The literature review in lines 18-27 needs supporting references.
Response 1: Thanks for your notice. The supporting references have been added.
Line 44: “…other work…” needs a reference. It is not clear here which research work you are referring.
Response 2: Done.
Lines 75-90 needs references. Otherwise, discussions are not supported
Response 3: Done.
Line 110: needs to define DNNs stands for what?
Response 4: Done. In addition, DNN has been included in the abbreviation list at the end of our paper.
Overall, authors needs to carefully edit their manuscript there are some errors in text, for example line 140, 320,…
Response 5: We have carefully checked and edited the paper once again. Thanks for your inspection.
Reviewer 3 Report
The authors propose the optimization of a pruned network using simulated annealing.
The proposed approach is compared with two simpler pruning heuristics.
The state of the art is incomplete. Please consider recent publications using SA for pruning.
The problem should be better formulated. As far as I understand, the approach just moves branches and does not remove or add branches. Is it? Also, the idea is to improve the structure of pruning. This should be clarified. What kind of structure are you searching for? How will it improve the final inference execution? Do you start with a pruned network or with a dense network?
Your network model has only 48 hidden units. This is a very small network and only considers fully connected layers. Other layers should be considered and larger models. Does the objective function consider the structure of the pruning?
Please indicate the execution times of the algorithms. Also, how long would it take to run fine-tuning training?
The objective is to avoid retraining. So, you should compare the execution times and accuracy of the pruned model with a solution after retraining. The compared heuristics are much simpler and faster than using SA. It was expected that the solutions obtained with these heuristics were worse.
How long does your approach take to prune larger models, like VGG or ResNet? Is it feasible?
The results presented in figure 6 are meaningless because you have an accuracy below 50%. You should consider a better model before pruning.
Author Response
The proposed approach is compared with two simpler pruning heuristics.
Response 1: Our proposed approach is compared with both uniformly random pruning and deterministic weight magnitude thresholding pruning strategies, which is not two heuristics.
The state of the art is incomplete. Please consider recent publications using SA for pruning.
Response 2: According to our study, the recent publications using annealing algorithm focused on helping the training process of a network without considering its configuration optimization. An additional paragraph has been added in the section of "Related Works", introducing the SA related methodologies proposed recently.
The problem should be better formulated. As far as I understand, the approach just moves branches and does not remove or add branches. Is it? Also, the idea is to improve the structure of pruning. This should be clarified. What kind of structure are you searching for? How will it improve the final inference execution? Do you start with a pruned network or with a dense network?
Response 3: The objective of our approach is to find the optimal structure of a network for a given percentage of pruning. This concept has been formulated in Equation 5. The pruning process starts from a pretrained fully connected (dense) network, which is explained in chapter 3.6. Under a fixed pruning percentage, the edges in a fully connected network will be removed first, and the remained edges will be permuted to optimal positions. The structure that we are searching for is the one that can obtain minimum loss value. After finishing the pruning process, the final inference execution can be done with far less weight parameters but still holding the same performance.
Your network model has only 48 hidden units. This is a very small network and only considers fully connected layers. Other layers should be considered and larger models. Does the objective function consider the structure of the pruning?
Response 4: The proposed objective function (Eq.5) does include the structure optimization, which is to find the optimal configuration represented by mask M. We study a modest size of a network to better understand the mechanism between weight parameters and network performance. Once it is clear, our algorithm can be better expanded to bigger networks.
For the words mentioned in the last question “the structure of the pruning”, we wonder whether you mean if some penalty terms are included for structure dimension? We haven’t considered introducing a penalty for network dimension yet. We agree that this is an important feature to include in the optimization problem.
Please indicate the execution times of the algorithms. Also, how long would it take to run fine-tuning training?
Response 5: Our approach is designed for the scenario that the computational power of edge devices is not enough to train a deep neural network with back propagation since gradients computation for each one of weight parameters will consume huge time and memory. As the training batch size is increased, the memory consumption will also increase linearly. On the other hand, Simulated Annealing can be done by simply doing inference without computing the gradients, which save huge memory space on edge devices. If we want to further compress a network on an edge device, our approach provides a path to prune weight parameters without further training it.
The objective is to avoid retraining. So, you should compare the execution times and accuracy of the pruned model with a solution after retraining. The compared heuristics are much simpler and faster than using SA. It was expected that the solutions obtained with these heuristics were worse.
Response 6: The reason why retraining is avoided is because of the hardware limitation. Therefore, we only focus on the performance loss between before and after the pruning process. Even though the compared strategies are much simpler and faster, we not only concentrate on the accuracy, but also care about the causation (the distribution of weight magnitude) that leads to the corresponding performance.
How long does your approach take to prune larger models, like VGG or ResNet? Is it feasible?
Response 7: We guess you ask about how the algorithm scales for high dimension networks. Our implementation of pruning process on a deep neural network (DNN) is done hierarchically layer-by-layer. If there are more layers, the time complexity will increase linearly. If there are more nodes contained in one layer, the time complexity will also increase linearly.
The results presented in figure 6 are meaningless because you have an accuracy below 50%. You should consider a better model before pruning.
Response 8: The reason why we prune a low performance network is because we want to show that the effect of our algorithm works independent to the network accuracy. It is proved that the performance degradation can be postponed by applying our algorithm.
Reviewer 4 Report
This paper proposes a new method for the Neural Network Structure Optimization by Simulated Annealing. The proposed method and the experimental results are clearly presented. In general, the quality of the paper is up to standard. However, in order to further improve the quality of the paper, I have the following comments. 1. In section 2 "Related work", the author can include a comparison for the proposed method with the past research work. A table could be used to summarize the relative merits and shortcomings of different methods. 2. It has been found that there are many past research work for the metaheuristic optimization algorithms, it may be useful if the author can describe more about the reasons why Simulated Annealing should be used for ANN structure optimization, instead of other metaheuristic algorithms (e.g. PSO Particle Swarm Optimization etc.). 3. The discussion/ conclusion in section 5 could be further extended to illustrate the original contribution of the proposed method.Author Response
1. In section 2 "Related work", the author can include a comparison for the proposed method with the past research work. A table could be used to summarize the relative merits and shortcomings of different methods.
Response 1: Thanks for the suggestion. The recently proposed methods have been included in section 2 “Related Works”. Although Simulated Annealing has been applied to deep neural network before, other researchers only focused on optimizing the branch weights instead of finding a better network configuration. We believe that since these approaches have different objectives, listing them in a table would not deliver the right message to the reader.
2. It has been found that there are many past research work for the metaheuristic optimization algorithms, it may be useful if the author can describe more about the reasons why Simulated Annealing should be used for ANN structure optimization, instead of other metaheuristic algorithms (e.g. PSO Particle Swarm Optimization etc.).
Response 2: PSO, Genetic algorithms and SA are all stochastic algorithms, which search for an optimal solution by interatively improving the solution via intelligent searching strategies. The main difference between these methods is the theory that support their convergence. SA is guaranteed to get the global optimal solution with the condition of detailed balance following the Markov Chain theory. On the otherhand PSO and GA risk falling into local minima. Since the review arrived 6 days before deadline, we did not have enough time to code and simulate these algorithms to provide numerical validation; however, we believe that the theoretical advantage of SA is clear.
3. The discussion/ conclusion in section 5 could be further extended to illustrate the original contribution of the proposed method.
Response 3: Done.
Round 2
Reviewer 3 Report
You are replacing pruning based on the traditional SGD with simulated annealing and without fine-tuning. Therefore, you should compare the accuracy and the computational complexity of both approaches for different model sizes.
You say that retraining is avoided because of hardware limitations. However, you are not training or retraining with the embedded computing platform. What hardware limitation are you referring to?
You must consider known models, apply your method and compare them in terms of accuracy, training time, memory, etc. They should be compared with those two pruning strategies and with SGD with pruning.
Author Response
You are replacing pruning based on the traditional SGD with simulated annealing and without fine-tuning. Therefore, you should compare the accuracy and the computational complexity of both approaches for different model sizes.
Response 1: We replace pruning based on the traditional SGD with SA without fine-tuning because this makes the pruning process possible to be executed on edge devices without sending the whole network back to a big server where it is originally trained. This is the motivation of our work. Therefore, we do not see much sense in comparing with SGD based weight training. These are not two alternative approaches. We optimize the neural network structure, when weight training is not feasible. SGD training of weights is about weight optimization which is not the topic of our article. Including such a comparison would confuse the reader and undo our motivation for this research.
As for the computational complexity, there are two aspects to it: which are space and time complexity. The traditional SGD based training that is back propagation has high space complexity. Namely, it needs high hardware requirements due to the gradient calculation and storage. On the other hand, SA based structure optimization has less space complexity, less complicated operations, less storage, but higher time complexity since progresses over a Markov chain which needs to be executed sequentially.
You say that retraining is avoided because of hardware limitations. However, you are not training or retraining with the embedded computing platform. What hardware limitation are you referring to?
Response 2: We are not retraining the model during the whole pruning process. We prune some edges and permute the network configuration with the strategy followed by Simulated Annealing. Since gradient needs to be computed with huge amount of RAM especially when batch size is large, training process should be executed on a powerful computer with abundant RAM. This is what we mean by the hardware limitation. Since our pruning method does not train a network with gradient descent, the whole pruning process can be executed on an edge computing platform.
You must consider known models, apply your method and compare them in terms of accuracy, training time, memory, etc. They should be compared with those two pruning strategies and with SGD with pruning.
Response 3: We guess by “known models” the reviewer means large network architecture such as VGG or ResNet. The main idea of this paper is to illustrate 1) pruning by thresholding is not enough in itself and needs to be followed by structure optimization to find the best configuration of the network 2) that SA is an efficient method for searching for the globally optimal structure. These two objectives are demonstrated in a small network to make the dynamics of the method clearer. These two results do not change in the case of bigger dimension networks.
The effectiveness of our network pruning approach is based on the theory of Simulated Annealing. Therefore, in order to avoid the potential interference to the experimental results, we decide to start with a simple network.
Round 3
Reviewer 3 Report
You must validate your approach about two aspects: accuracy and performance.
To validate the accuracy, you must compare it with something. Pruning without retraining is a valid method but you must assess its quality and this should be done against a method that can potentially achieve better results, and this is retraining with pruning.
Section 4.3 is about the execution time of the algorithm. However, you did not specify the platform you are using to run the algorithm. What edge computing platform are you using?
About the performance of the algorithm, you must show its scalability or determine its limitation in terms of model size. You obtain up to 150 minutes to run small models. Given the more than linear increase in complexity of SA, do you consider it feasible to run the method for larger models?
When you say that SA is efficient for searching globally optimal structures, the efficiency is somehow determined by the model size. That is why you should characterize the scalability of your solution for larger models.
Author Response
To validate the accuracy, you must compare it with something. Pruning without retraining is a valid method but you must assess its quality and this should be done against a method that can potentially achieve better results, and this is retraining with pruning.
Response 1: I think we have not been able to express our replies well in the last round. We compare the performance in the case of network optimization after pruning (which is our proposal) with the case of no pruning. Since we assumed that the algorithm can only consume restricted computing resources on edge devices, those methods that consume more than the restricted resources are not considered in our experimental results. Including back propagation into the problem would remove the motivation of this work.
Section 4.3 is about the execution time of the algorithm. However, you did not specify the platform you are using to run the algorithm. What edge computing platform are you using?
Response 2: We simulate an environment of edge device on a regular windows computer by restricting the usage of RAM and threads without actually putting the algorithm into the real device. The objective edge computing device that we simulated is the Jetson series produced by Nvidia company. Jetson device contains 4GB RAM and 32GB storage space. In our experiment when SA is applied to optimize the network structure, it consumes less than 3GB RAM, which is well below the limit of the Jetson edge device.
About the performance of the algorithm, you must show its scalability or determine its limitation in terms of model size. You obtain up to 150 minutes to run small models. Given the more than linear increase in complexity of SA, do you consider it feasible to run the method for larger models?
Response 3: As studied in Figure 8, the idea choice of metropolis loop length (MLL) is somewhere between 20-50. The difference in accuracy performance between 20 and 50 MLL is less then 1%. Then, one can safely choose MLL to be 20 which takes 35 minutes. In our work, the objective is to show the importance of network configuration by the experimental results with the help of SA optimization algorithm. Based on the conclusion, the problem of scalability will be our future work to focus on. The time consumption of 150 minutes is the extreme case where the Metropolis loop length (MLL) equals to 100. According to our experimental results, the best MLL locates around 20 to 30.
As for the application to the larger models, this issue has been described in the last paragraph of discussion section. In this work the smallest structural unit we considered is an edge, however for larger architectures we can consider the structural unit to be a node, a filter, or kernel (in CNN), which will significantly reduce the search space. Therefore, the increase can reduced to sub-linear. The evaluation of the time consumption under different edge numbers and different Metropolis loop length have been compared in Sec 4.3 of the latest revised paper.
When you say that SA is efficient for searching globally optimal structures, the efficiency is somehow determined by the model size. That is why you should characterize the scalability of your solution for larger models.
Response 4: The effectiveness of determining global optimal structure is compared with the case of naïve grid search. As the total number of edges increased, there is exponentially more combinations to represent network structures. However, the computational complexity of SA increases linearly .